# University Students' Perception, Evaluation, and Spaces of Distance Learning during the COVID-19 Pandemic in Austria: What Can We Learn for Post-Pandemic Educational Futures?

**Tabea Bork-Hüffer** [1] 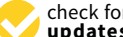, **Vanessa Kulcar** [2] , **Ferdinand Brielmair** [1] , **Andrea Markl** [1] , **Daniel Marian Immer** [2] , **Barbara Juen** [2] , **Maria Hildegard Walter** [2] and **Katja Kaufmann** [1,3,*]

1 Institute of Geography, University of Innsbruck, 6020 Innsbruck, Austria; tabea.bork-hueffer@uibk.ac.at (T.B.-H.); Ferdinand.Brielmair@student.uibk.ac.at (F.B.); Andrea.Markl@uibk.ac.at (A.M.)
2 Institute of Psychology, University of Innsbruck, 6020 Innsbruck, Austria; Vanessa.Kulcar@uibk.ac.at (V.K.); Daniel.Immer@uibk.ac.at (D.M.I.); Barbara.Juen@uibk.ac.at (B.J.); Maria.Walter@uibk.ac.at (M.H.W.)
3 Institute for Comparative Media and Communication Studies, Austrian Academy of Sciences and University of Klagenfurt, 1010 Vienna, Austria
* Correspondence: Katja.Kaufmann@uibk.ac.at

**Abstract:** The COVID-19 pandemic caught societies worldwide unprepared in 2020. In Austria, after a lockdown was decreed on 16 March 2020, educational institutions had to switch to a patched-up distance learning approach, which has been largely maintained to date. This article delivers empirical insights from an interdisciplinary mixed-methods research study that investigated university students' perceptions of and experiences with distance learning as well as their educational (home) spaces during the pandemic in Innsbruck, Austria. It combines results from a quantitative survey conducted with 2742 students in early 2021 with a qualitative multi-method and longitudinal research study that accompanied 98 students throughout four data-collection phases in 2020. Results show a significant improvement since spring 2020 with both teachers and learners adjusting to the distance learning formats and the use of digital tools, yet students urgently desired a return to face-to-face teaching and university life, particularly for its social benefits. Strikingly, more than half of the participants wanted to maintain the option of overall distance education after the pandemic. Based on the perspectives of students, it is appropriate to demand significant changes in post-pandemic education adapted to the era of the post-digital, for which this article gives short-term as well as medium-term recommendations.

**Keywords:** COVID-19; young adults; hybrid learning; remote teaching; educational spaces; tertiary education; higher education; Austria; mixed methods; post-digital

## 1. Introduction

Worldwide, 90% of all pupils and students (1.5 billion) were allegedly affected by the closure of educational institutions effected as a response to stall the spread of the COVID-19 pandemic in 2020 [1]. Austria, too, was affected by wide-reaching restrictions imposed early-on on educational institutions, including the province of Tyrol, home of the ski resort of Ischgl, which later rose to fame as a pan-European COVID-19 hotspot. Between mid-March 2020 and March 2021, overall three nationwide so-called "strict lockdowns" (1st lockdown from 16 March till April 2020, 2nd lockdown from 17 November till 6 December 2020, 3rd lockdown from 26 December till 7 February) were enacted, which encompassed stay-at-home orders, social distancing measures, and the extensive closure of non-essential businesses, in order to contain the spread of SARS-CoV-2. Higher education in Austria switched to distance learning starting with the first lockdown in March 2020. Since then, the tertiary educational institutions have not returned to in-person forms of teaching. Besides for slight relaxations during summer 2020 when infection numbers had dropped

significantly, distance learning was maintained until the time of writing this article in June 2021.

In this article, we present results from a comprehensive interdisciplinary mixed-methods research study, which investigates the complexity of perceptions, evaluations, practices, experiences, and spatial contexts of distance learning for students in the Tyrolean capital of Innsbruck in Austria throughout the first year of the COVID-19 pandemic. The article focuses on empirical answers to the following research questions: how did students perceive and evaluate distance learning to date? Which spatial—i.e., social, material and digital—contexts have influenced distance learning? What are the lessons learned for the short-term design of distance education during the pandemic and a mid-term provision of post-pandemic education adapted to the era of the post-digital [2]? The data collected with students on distance learning during the COVID-19 pandemic that we use to answer these questions were collected as part of a mixed-methods research project collaboratively conducted by the Institutes of Geography and Psychology at the University of Innsbruck in Tyrol. The study applied a sequential design in which the qualitative results from a longitudinal qualitative multi-method study that included 98 university students were used to design a follow-up quantitative survey with 2742 students from the University of Innsbruck.

After introducing existing research and our methodology, we present the background to the understanding of the empirical results regarding the types of courses visited and the digital media used for distance learning during the pandemic. Next, we show the results of the study regarding students' perceptions, evaluation of distance learning, the socio-material home spaces in which they pursued distance learning, their social media use, and the effects of these variables on distance learning. Eventually, we discuss recommendations for pandemic education and for distance education beyond the pandemic that is more socially sustainable and responsive to the realities of the post-digital.

## 2. Effects of the COVID-19 Pandemic on the Spatiality of University Students' Education and Learning

The worldwide closure of educational institutions due to the COVID-19 pandemic sparked interest among many researchers in how the forced conversion to distance education and learning affected teachers and learners. Indeed, numerous national and international studies were conducted in 2020 that examined various aspects of the transition to and implementation of distance education and learning. Many of the studies explore either academic work and life [3–5] or students' mental health [6–11], or a combination of the two topics [12,13].

Comparative studies on the psychological effects of the pandemic suggest that young people, and particularly young adults, are especially vulnerable to the mental health effects of the pandemic [14,15]. This vulnerability partly stems from their developmental phase. Young adults in their qualification periods are in a formative transition phase of their lives, which is marked by insecurities and changes [16,17], even more so with a pandemic around [18]. The additional challenges during the COVID-19 crisis can therefore easily overwhelm them. As their studies, and therefore also the universities, play a central role in students' lives, changes in how the education system works affect all other aspects of their everyday experiences. Social contacts among students decrease [19], and some move back in with their parents [20] or find themselves separated from families and friends while studying abroad due to closed borders [21]. In addition, they are under pressure to keep up their performance while having difficulties finding motivation and concentrating during the time of crisis [22]. As the young adults' lives profoundly change and their mental health is severely affected, various studies suggest that the mid-term consequences will likely be felt for some years after a withdrawal of pandemic-related measures [23–26].

Furthermore, current research sheds light on the dimension of sustainability in the context of the pandemic. Studies on sustainability education [27,28] as well as on sustainable online learning and teaching [29–31] have highlighted challenges and possibilities for sustainable development in the context of the pandemic. Especially social distancing and

temporary lockdowns are reported to have a significant impact on the everyday life and education of millions of students. Sustainable online learning, which includes access to and the possibility of actively participating in online education during the pandemic, is considered to be a decisive factor for successful teaching and learning and, furthermore, a key factor for maintaining and increasing social sustainability. Social sustainability is therefore at risk if students have limited access, or even no access at all, to online education.

Lastly, economic impacts play a significant role, and they directly affect tertiary education institutions. In February 2021, 436,982 people were registered as unemployed with the Austrian public employment agency AMS [32] (p. 1), and a further 71,941 participated in training courses [32] (p. 3), out of a total Austrian population of almost 9 million [33]. The cohort of those aged 15–24 years was already more affected by unemployment before the crisis and remains one of the most affected groups with a lack of prospects on the job market. As a result, student enrolment, including in the tertiary education institutions of Tyrol, is increasing. Meanwhile, students already enrolled feel that their performance has been dropping due to the mental-health impacts of the crisis [22]. Universities are challenged to accommodate these different groups and interests and at the same time need to care for students' mental health to avoid negative long-term effects—all while building up and maintaining a teaching system almost completely based on distance learning.

In studies on students' experiences of the COVID-19 pandemic, the spatiality of education and learning, however, is hardly considered. This should come as a surprise as the COVID-19 crisis has a very palpable spatial dimension. Distance learning per se implies spatiality, both through the relocation of aspects of learning and teaching to the students' home spaces and their entanglements with digital learning spaces, and through the fact that students no longer experience circulation between their home spaces and the university as a space of education due to this outsourcing. Gobbi and Rovea [34] argue, in this context, that spatial changes due to distance learning go hand in hand with a reduction of the university as a joint (inter)actional and socio-material space to scattered digital spaces on one or more devices. These devices are embedded in the everyday living environments, the home spaces, of the students [35], which is why students experience a new kind of homemaking. Furthermore, while the home space, with various functionalities already intersecting in it, becomes the (new) individual educational and learning space, this does not mean that students find themselves alone in this new assemblage. Rather, they are embedded in altered relational networks [36], in which people, objects, and routines remain closely connected [35] and sociality, including the collective of the university, is maintained by digital means [37]. Following Bork-Hüffer et al. [38], one can speak of cON/FFlating educational and learning spaces that are simultaneously shaped by entangled socio-material and techno-social relations and spaces. They remain socially shaped by multiple perceptions, meanings, values, and ideologies of students, teachers [36], and peer-groups. In line with this, the entangled material, social, and technological dimensions of educational and learning spaces must be taken account of in the analysis of distance learning and its impact on students.

The importance of space to education and learning was already emphasized as part of the spatial turn in the geographies of education [39]. Without a doubt, during the pandemic, new and altered formal and informal educational and learning spaces are emerging in which learners in turn have learning experiences that are particularly central to the social and cultural geographies of education. Given the importance of the affective-emotional experience of space, i.e., of (mediated) sense of place [40–42], it can be argued that parallel to the (new) educational and learning spaces, new manifestations of a sense of educational and learning place also emerge. Subsequently, these new spatially shaped learning experiences condition a constant reproduction and interpretation of students' identities through socio-spatial-technological practices [36], which is also inscribed in the formation of a sense of educational and learning place [43,44]. Student identity, or a student identity specific to each individual, should also be interpreted in the context of the living situation, the home space, because it can influence students' experiences as such both

positively and negatively and has an impact on identity formation within social activities, learning environments, friendship networks, and other socio-cultural factors [45].

## 3. Methods

The data that are the basis of the analysis presented in this article were collected as part of a sequential mixed-methods research design, in which the insights from a qualitative multi-method project (study part 1) were used to design a quantitative survey instrument (study part 2). While the data collection was sequential, an integrated data analysis followed in which both forms of data were merged.

Part 1, the qualitative part of the study, built upon an analysis of data collected as part of the ongoing research project COV-IDENTITIES at the Institute of Geography of the University of Innsbruck, Austria. The project applies a qualitative multi-method and longitudinal approach, accompanying young adults, among them university students, since early April 2020 throughout the COVID-19 pandemic. The aim of the COV-IDENTITIES project is to explore changes in young adults' socio-material and techno-social everyday spaces and practices (including study, work, contacts, leisure) over the course and different phases of the COVID-19 pandemic. The COV-IDENTITIES dataset includes comprehensive data related to distance learning. It combines written narratives with smartphone-based methods (mobile instant messaging interviews) [46] and in-depth interviews.

Written narratives offer a qualitative approach to elicit individual experiences with and reflections on complex change processes as well as potentially traumatic experiences [47]. The approach gives young people room for subjective descriptions and interpretations of their feelings and experiences [48,49]. Mobile instant messaging interviews (MIMIs) offer comprehensive in situ insights into the very concrete daily practices and spaces of selected participants [50]. Using established messenger apps (here WhatsApp), research participants are contacted by researchers at regular (in our case, two-hourly) intervals and are asked about their current practices and spaces, and also about socio-material and techno-social elements with which they engage during those practices, against the backdrop of their daily routines. In contrast to standardized mobile experience-sampling approaches, when using MIMIs, researchers engage with participants in the digital instant messaging spaces where they explore together the participants' momentary experiences [46]. MIMIs thus allow researchers to gain deep insights into the complex everyday spaces and practices of research subjects in situ, even under conditions of curfews and social distancing. Research subjects can use a wide range of multimedia elements (text messages, photos, videos, voice messages, GIFs, emojis, screenshots) for their responses. To maximize the variety of perspectives in our qualitative study, additional participants from the pool of narrative data were invited to participate in in-depth interviews, to facilitate even more comprehensive insights into different perceptions, evaluations, experiences, and strategies with distance education.

Longitudinal data collection using narratives and MIMIs took place in four phases (DCs) in 2020:

- DC-1: 1–7 April: 1st lockdown
- DC-2: 20–27 April: during stepwise relaxation of the 1st lockdown
- DC-3: 2–14 June: full relaxation of the 1st lockdown
- DC-4: 16–23 November: 2nd lockdown.

The in-depth interviews were conducted in parallel in July, August, and November 2020. Data collection is planned to continue throughout 2021.

The sample encompassed students who were enrolled at universities in Innsbruck at the time of DC-1. The participants were aged between 18 and 36 years, and 65% were female (average at DC-1). As part of the COV-IDENTITIES project, 340 written narratives and 44 full-day MIMIs with university students were collected throughout the four DCs. The exact number of participants includes, for narratives in DC-1: $n = 98$, DC-2: $n = 93$, DC-3: $n = 82$, DC-4: $n = 67$; and for MIMIs, DC-1: $n = 13$; DC-2: $n = 12$, DC-3: $n = 8$, DC-4: $n = 11$. An additional ten follow-up in-depth interviews with students were conducted

that focused on distance learning only. The participants in the interviews were selected with the aim of collecting a maximum variety of perspectives, ranging from students who had previously reported strongly negative experiences with distance learning to those who had reported strongly positive ones.

The qualitative data were analyzed with the help of MaxQDA using qualitative content analysis, which is an appropriate strategy for analyzing data from all three methods (narratives, MIMIs, interviews), e.g., [47]. The WordCloud used in Section 4.5 was produced in MaxQDA (Berbi Software, Berlin, Germany) and shows all software (including distance learning software) that was mentioned in the written narratives in different sizes according to the number of times they were named in narratives.

Based on the insights gained from the qualitative part 1, in part 2, a quantitative survey was designed to expand the findings of the qualitative part, to be conducted with a larger population of students at the University of Innsbruck. This survey was based on the results of the qualitative data set, as well as on outcomes of previous surveys conducted by this workgroup, which were designed to elicit information about the mental health and practices of students during the COVID-19 crisis. The resulting questionnaire included questions about different criteria relevant to students' perceptions and experiences of distance learning. These included contextual factors, such as living conditions, spaces where students take part in distance learning, and technical equipment; personality factors, including motivation and the ability to work independently; and subjective workload and information on the concrete implementation of students' courses. Additionally, data about participants' satisfaction with distance learning and performance, measured as the number of exams taken and scores achieved, were retrieved. The survey was sent to students at the University of Innsbruck on behalf of the vice-rector of teaching and studies. For the students, this recruitment by university officials indicated the importance of the survey for the future shaping of their own studies, and thereby likely increased their motivation to participate and answer all questions truthfully. The survey was open for replies from 19 February to 8 March 2021. The survey was presented in a multiple-choice and Likert response format, and students could also choose not to answer individual questions, to avoid forced answers.

In total, 2742 students completed the survey, which equals 10% of the total number of students at the University of Innsbruck in 2020 [51]. Most participants were between 20 and 30 years old (79%). Females were slightly overrepresented, making up 66% of the survey's participants (whereas females constitute 53% of all students at the University of Innsbruck). Males made up 32.5%, while 1.5% did not assign themselves to either of these genders, or did not want to answer the question.

To explore students' satisfaction with distance learning, we used six items asking for their satisfaction with distance learning in general, the workload, the content, their learning success, and the university's communication. The items were rated on a 5-point Likert scale (1 = 'I do not agree at all' to 5 = 'I agree completely'). A scale was created by calculating the mean. The internal consistency was satisfactory (Cronbach's $\alpha = 0.82$). To explore students' subjective performance during the distance learning semester, we used three items asking about changes in the number of exams and ECTS compared to before the distance learning started. The items were rated on a three-point Likert scale ($-1$ = fewer exams/ECTS; 0 = no significant changes; 1 = more exams/ECTS). Cronbach's $\alpha = 0.88$ was satisfactory. We calculated the mean of both items, with values <0 indicating decreased performance and values >0 increased performance.

For this article, the quantitative data set was analyzed using IBM SPSS Statistics, Version 26 (IBM, Armonk, NY, USA). We conducted descriptive analyses, independent *t*-tests, analyses of variance, and correlation analyses. Participants had the opportunity to select "no answer", so sample sizes varied for different tests. Effects were interpreted as significant with $p < 0.05$. We used Bonferroni correction when we ran multiple tests to avoid error inflation.

Eventually, an integrated analysis stage took place in which both data sets were analyzed with regard to findings for the main research questions presented above, and findings in the quantitative survey were cross-checked with the qualitative data and vice versa. Fully building upon the advantage of the mixed-methods study, the results are closely interwoven in the following results section. All data are used there to give substantiated and broad insight into students' perceptions, evaluations, and spaces of distance learning during the COVID-19 pandemic.

## 4. Results

### 4.1. Types of Courses Attended and Types of Distance-Learning Practices

Figure 1 summarizes the types of courses that the students reported having attended during the winter term, which lasted from 1 October 2020 till 1 February 2021. Most of the courses were teacher- and classroom-focused formats (lectures (VO, Vorlesungen), lectures with exercises (VU, Vorlesungen mit Übungen), proseminars (PS, Proseminare), seminars (SE, Seminare), or orientation courses (SL, Studienorientierungslehrveranstaltungen)), but a considerable percentage also attended classes that include practical, field, or laboratory experiences in normal, non-pandemic study terms (exercise courses (UE, Übungen), field trips (EX, Exkursionen), field courses (EU, Exkursionen mit Übungen), practical training sessions (PR, Praktika), work groups (AG, Arbeitsgemeinschaften)). Overall, 35% of the students reported in the quantitative survey that they had taken part in at least one course that took place in a physical, face-to-face format during the winter term 2020/2021.

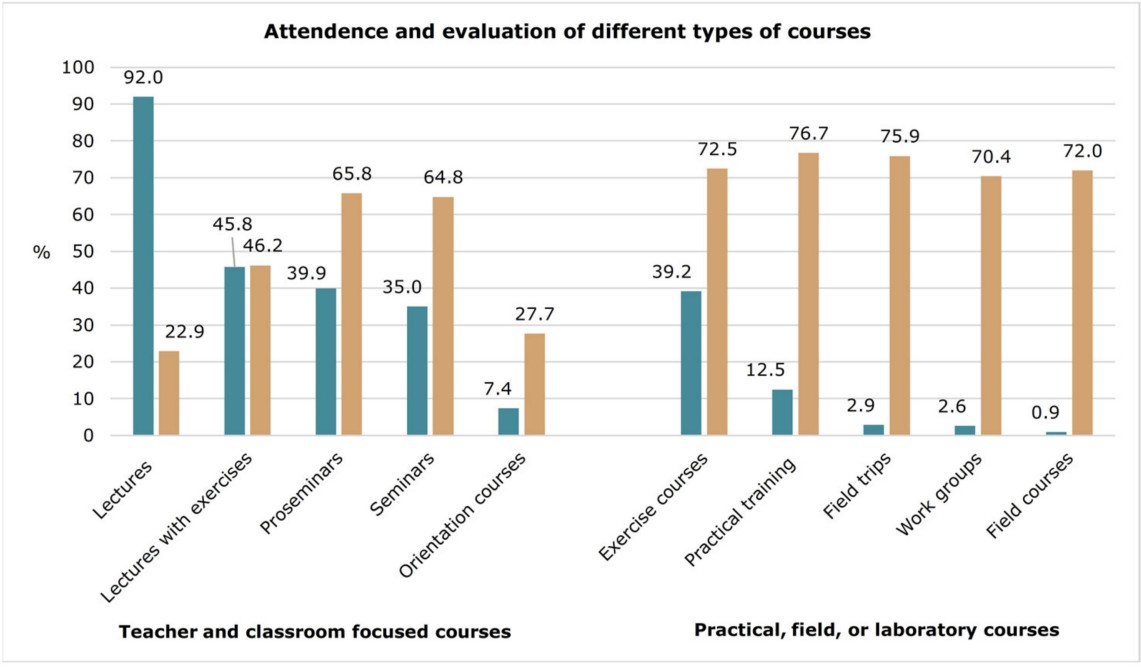

**Figure 1.** Types of courses that students attended during the winter term 2020/21 (green) and types of courses that suffered most from the conversion to distance learning (brown). Types of courses that suffered most are filtered by students taking the courses during the winter term 2020/21. Lectures *n* = 2524, lectures with exercises *n* = 1256, proseminars *n* = 1094, seminars *n* = 959, orientation courses *n* = 202, exercise courses *n* = 1075, practical training sessions *n* = 343, field trips *n* = 79, work groups *n* = 71, field courses *n* = 25 (multiple choice, source: quantitative survey, February 2021).

During the pandemic, the tasks pursued most often during distance learning, i.e., following lectures online or working on tasks given independently at home, did not include direct teacher–student interaction, as Figure 2 shows. This is generally not surprising, as a large part of learning tasks for students involves listening to lectures or self-study even during regular (non-pandemic) study terms. However, the use of online forums for teacher–student exchange and live question–answer sessions were new or at least largely increased

types of interaction that sprung up during the pandemic. Furthermore, 34.9% reported that during the winter semester 2020/2021 they had worked in parts of their courses in small groups, and 21.8% that they had done so in large groups, indicating that interactive formats were maintained in some course types.

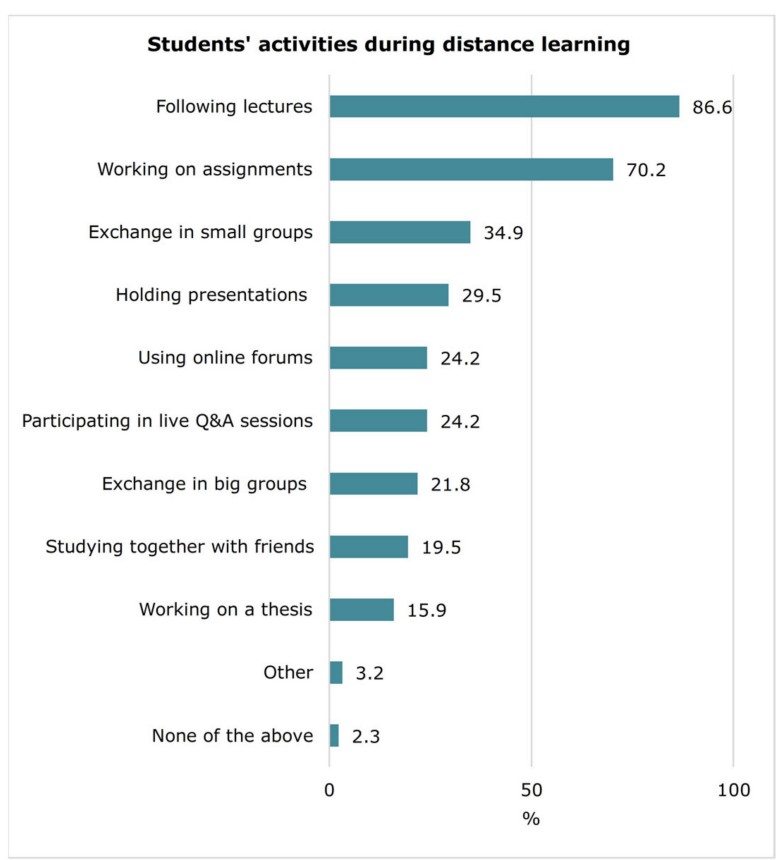

**Figure 2.** Types of activities that students conducted during the winter term 2020/2021 (multiple choice, source: quantitative survey, February 2021).

### 4.2. Hardware and Software Used for Distance Learning

To access the Internet, as the most important link between teachers and students, as well as in their learning and working process during distance learning, the students used various types of hardware. They primarily used their laptops/computers, smartphones, e-book readers, game consoles, and televisions for this purpose. Laptops and stand-alone PCs were used, as they were before the pandemic began, for word processing; running, applying, and learning course-relevant and helpful software; research; and communication. New additions to distance education included attending classes via online (learning) platforms and videoconferencing software, as well as accessing and watching instructional videos created by instructors for learners and posted online. Students used their smartphones in addition to their computers either alternately or simultaneously. These were often used for personal purposes, but they were also used in distance learning. Students used their smartphones to communicate with instructors and other students via various messenger services and email apps. However, smartphones were also used to retrieve and watch instructional videos and to participate in video conferences in virtual classrooms. In some cases, apps were used to help students organize and complete work assignments and to gather information.

Some students also indicated that they used their TVs to attend courses (primarily lectures) and to watch instructional videos on a larger screen and in a more comfortable setting, such as on the couch at home. Game consoles were also used, providing a range of software that students used both to communicate and to access instructional videos.

Due to the closure of libraries and the resulting limited access to literature, some students also used their e-book readers for research purposes and to read course-relevant literature, provided that the relevant literature was also available in digital form.

In addition to the hardware, students mentioned a variety of software they used during the COVID-19 pandemic. Figure 3 provides an overview of the software, categorized by their primary function in distance education. It should be noted here that in addition to the software used across the university, students also used specialized software specific to their discipline (e.g., ArcGis, MATLAB, SPSS). Challenges particularly related to the use and availability of specialized software are explored in more detail in Section 4.3.

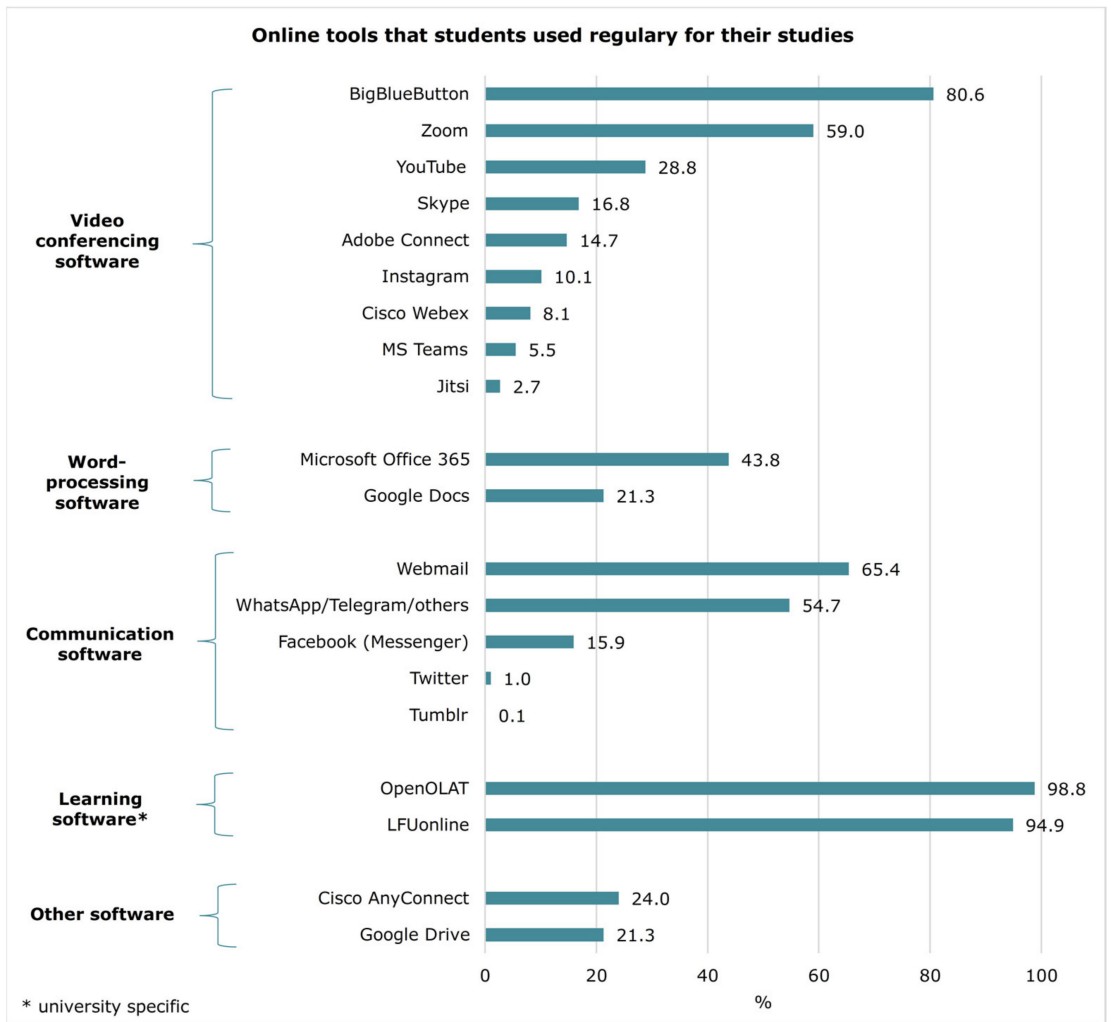

**Figure 3.** Software used in distance education across institutions (multiple choice; source: quantitative survey, February 2021).

As can be seen from Figure 3 showing the variety of software used across the university, students used a number of different kinds of video (conferencing) software, most of which was chosen by instructors for distance learning. It should be mentioned here that based on the insights from the qualitative longitudinal part, we can conclude that in the summer term 2020, at the beginning of the COVID-19 pandemic, many students reported a higher number of different kinds of video (conferencing) software used in distance learning than in the following winter semester 2020/21. With regard to word-processing software, some students stated that especially the possibility of simultaneous online editing of text files proved to be helpful while working on group tasks. For communication within the group, students used social-media software such as the Facebook and WhatsApp platforms and

messenger services, as well as e-mail software such as Webmail. In addition to Webmail, students also used the (learning) software of Innsbruck's universities to communicate with their instructors, conduct research, and access information about course content and organization.

Overall, students generally used digital media more frequently and over a longer period of time since the switch to distance learning (see also Section 4.3). A more detailed description of the challenges with digital media follows in the next subsection.

### 4.3. Experiences with and Evaluation of Distance Learning during the Pandemic

Students had very different experiences in connection with the conversion to distance learning. Overall, the type of course chosen clearly had an impact on the perception of distance learning, as Figure 1 shows. The courses that contained practical elements were considered to be the most negatively affected by the conversion to a distance learning format. Still, students taking practical, field, or laboratory courses during the winter term 2020/21 did not differ from students not taking such courses in their satisfaction with distance learning ($t(2688) = 1.83$, $p = 0.067$). Additionally, there were no differences in performance ($t(2193) = -0.54$, $p = 0.589$).

The qualitative study highlights major individual differences, for example in the perceived challenges, individual structural conditions of distance learning (see also Section 4.4), attitudes toward distance learning, and in the personal appropriation of digital media (see also Section 4.5). For the purpose of comparison, the students' experiences are reduced and categorized in the following to challenges (negative experiences) and potentials (positive experiences) of distance learning. It should be noted that the qualitative data made clear that some elements of a course could have been perceived positively and other elements negatively. Figure 4 quantifies the perception of elements of distance learning based on the results of the quantitative survey.

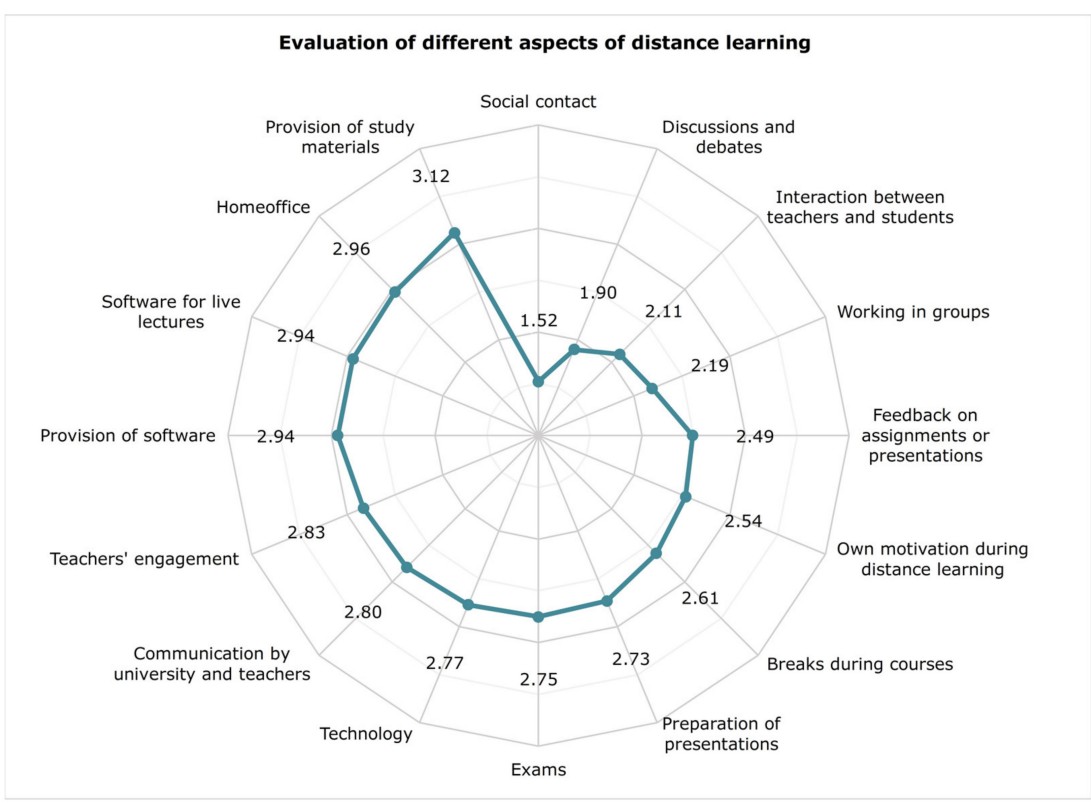

**Figure 4.** Evaluation of social, communicational, organizational, technical, and personal dimensions of distance learning ("How well do the following aspects work in distance learning in general?", single choice, 1 = 'very poorly', 2 = 'poorly', 3 = 'well', 4 = 'very well', source: quantitative survey, February 2021).

### 4.3.1. Negative Experiences: Challenges of Distance Learning

In the quantitative and qualitative data, students reported negative experiences with distance education, citing a variety of reasons why they were dissatisfied with distance education in different situations. One aspect that resonated through most responses was the lack of social exchange and contact in the context of distance education. Students missed the interaction with instructors (74.4%) and fellow students (91.9%) during class, as well as casual interactions (85.5%) (see also Figure 4). Those who missed these interactions reported decreased satisfaction with distance learning and decreased performance (see Table 1).

**Table 1.** Group differences between students missing interactions during distance learning and students not missing them.

| | *n* | **M** | **SD** | *n* | **M** | **SD** | **T** | **d** | *p* |
|---|---|---|---|---|---|---|---|---|---|
| | | **Missing** | | | **Not Missing** | | | | |
| **Satisfaction with Distance Learning** | | | | | | | | | |
| Interactions with instructors | 2017 | 3.32 | 0.80 | 695 | 3.84 | 0.76 | 14.91 | 0.66 | <0.001 |
| Interactions with other students | 2491 | 3.40 | 0.80 | 221 | 4.05 | 0.78 | 11.54 | 0.81 | <0.001 |
| Casual interactions | 2318 | 3.41 | 0.81 | 394 | 3.74 | 0.81 | 7.55 | 0.41 | <0.001 |
| **Performance during Distance Learning** | | | | | | | | | |
| Interactions with instructors | 1623 | −0.11 | 0.58 | 588 | 0.11 | 0.60 | 7.82 | 0.38 | <0.001 |
| Interactions with other students | 2011 | −0.07 | 0.59 | 200 | 0.18 | 0.61 | 5.52 | 0.42 | <0.001 |
| Casual interactions | 1883 | −0.07 | 0.59 | 328 | 0.07 | 0.61 | 3.92 | 0.24 | <0.001 |

Note. *n* = sample number for each response option, M = mean, SD = standard deviation, T = *t*-test, d = Cohen's d, *p* = significance level.

Many found that social exchange, which is required and encouraged in face-to-face teaching, is more difficult in online formats. Debates and discussions during class were missed or perceived as poor by a considerable number of the students (78.0%). Group and team assignments and projects were perceived as more difficult in distance education than in face-to-face education. Participant Emma (24, narrative, DC-2), for example, explains that *"an assignment [is] significantly more time-consuming online than it [would be] in the university, there is often a lack of exchange during group work."*

Furthermore, the anonymity, particularly during lectures and on online learning platforms, was criticized:

*"The most negative thing about distance learning, in my opinion, is the lack of contact with fellow students. Sure, you can [use] Skype and write emails, but I only do that with those I already know. In some courses I don't know anyone and since the webcams are never turned on, I don't have any faces to go with the names and this anonymity is quite strange and I feel quite uncomfortable because you can't make any contacts that way."*

(Annika, 24, narrative, DC-4)

With 37.1% evaluating feedback on their own work and presentations by teachers as poor or very poor, giving feedback during distance learning did not work well for a considerable share of the participants. The students also stated that the exchange with lecturers and fellow students before, after, and between the courses is important to them and has a significant influence on their well-being and potential learning effects. In the quantitative survey, 85.5% noted that they missed the informal opportunities for discussion and exchange outside the classroom sessions.

The results of the quantitative survey show that students were relatively satisfied with the engagement of teachers, their communication, and their provision of learning material as well as the overall communication by the University (see Figure 4). The qualitative longitudinal data indicate that this evaluation has strongly improved since the beginning of the pandemic. Especially in the weeks of the first lockdown in March and April of the summer semester 2020, during the acute changeover phase to distance learning (survey phases DC-1 and DC-2), students experienced difficulties due to lack of

information about courses, lack of (clear) communication from lecturers, lack of information about examination regulations and formats, especially from some lecturers, but also from institutes, examination offices, and the University:

> *23.04.20, 17:10—Interviewer: How would you describe your current emotional state?*
>
> *23.04.20, 17:13—Rebecca: Upset. I got an email from the internship office a few minutes ago that kills me. Everything is so uncertain in my studies and all we keep getting is empty promises and actually no real information. I'm extremely worried whether I'll be able to graduate this semester as planned and that just upsets me immensely [...] Actually, I'm slowly running out of energy to continue and still study because I just don't know what for and I'm not learning anything because we don't get any fixed exam dates.*

(Rebecca, 26, MIMI chat log, DC-2)

However, even after the first lockdown and a hasty conversion to distance learning, in the qualitative data, students continued to note the lack of availability of some teachers, the perceived lack of technical and didactic skills of some teachers, and the feeling that some teachers exerted too much pressure on students. This included, for example, the use of software that was unsuitable for the implementation of the respective teaching mode, an insufficient use of functional possibilities of software, and incomprehensible sound and video recordings. Students criticized an insufficient adaptation of the teaching concepts to digital circumstances in some courses, or the replacement of teaching by the sole provision of materials (PowerPoint slides or literature).

In addition, in the qualitative data collection, students provided data about the psychological impacts of distance learning, particularly regarding motivation and concentration problems, but also about a loss of joy in studying, increased stress, feelings of being overwhelmed, and depression. They noted that during courses without active involvement, their concentration decreased, and more so than in face-to-face teaching formats:

> *"I find long seminars too exhausting, because even with breaks my concentration in front of the laptop quickly drops, especially when I don't have anything to do actively"*

(Lotta, 23, narrative, DC-4)

In addition, some students reported feeling stressed by a perceived pressure to be constantly available online. However, the opposite effect—a lack of engagement and resulting decreased stress due to their physical distance from the educational institution of the university—was also mentioned by some students. Some of these students eventually raised concerns as to what the point of their study would be, when no intensive personal involvement with course content was necessary in order to successfully complete courses. Some students found the large amount of instructional and work material burdensome and, in some cases, overwhelming. While many students described familiarizing themselves with digital software as incidental and self-evident, previously unknown digital software, or the lack of opportunity to try it out prior to using it caused some study participants to feel stressed and anxious before presentations and exams. Feeling overwhelmed with materials in the distance learning situation and worrying about presentations or exams led some students to decide not to take exams, which prevented them from successfully completing some of their courses. A few students reported being severely overwhelmed and described having experienced depressive episodes during the pandemic.

In addition to the negative effects of distance learning on students' mental health, distance learning also had a negative impact on their physical health. For example, in the qualitative study, students communicated that they had suffered from headaches, back pain, sore eyes, and discomfort associated with working with digital devices. They also raised the issue of increased and quicker-onset of fatigue associated with screen-based work.

Moreover, participants in the qualitative data collection described a variety of technical difficulties, especially associated with the (internet) connection, software, and hardware. They reported connection problems and slow internet transmission rates, which interfered with smooth distance learning. Students with poor internet connection quality were both

less satisfied with distance learning ($r = 0.27$, $p < 0.001$, $n = 2712$) and reported reduced performance ($r = 0.13$, $p < 0.001$, $n = 2211$). Other students criticized the software used for distance learning and found it not to be functional or user-friendly. Some reported that they were distracted by the large amount of software, which, however, also included media used for purposes other than distance learning (see also Section 4.5). Due to the high dependency on software, technical failures were all the more detrimental to productive distance learning. Problems with microphones were found to be particularly disruptive, as microphones provided the only opportunity for direct verbal exchange. Since some (special) software only works on suitable hardware, additional problems arose for some students.

In the quantitative survey, 38.1% reported that they were forced to purchase new hardware for distance learning purposes. The reason given for this was that most laptops are not designed for computing energy-intensive (special) programs and the university did not grant access to the devices located within the university buildings for reasons of COVID-19 protection measures. Funding and obtaining suitable hardware presented a challenge for some students, and they described relying on their parents for financial and material support because of this. Participants who had to make additional purchases for distance learning were less satisfied (M = 3.25, SD = 0.81, $n = 1030$, $t(2710) = -10.62$, $p < 0.001$, d = 0.41) and reported reduced performance (M = $-0.12$, SD = 0.63, $n = 867$, $t(1722.32) = -4.23$, $p < 0.001$, d = 0.20) compared with students not having to make such purchases (satisfaction M = 3.58, SD = 0.81, $n = 1682$; performance M = $-0.00$, SD = 0.57, $n = 1344$). Others reported that they felt compelled to disregard COVID-19 conditional spacing rules and collaborate (sometimes in groups) on an available powerful computer because a new hardware purchase was not financially economical and/or possible.

### 4.3.2. Positive Experiences: Potentials of Distance Learning

Both in the qualitative and quantitative study, students described various positive elements and circumstances in the context of distance learning (see Figure 5).

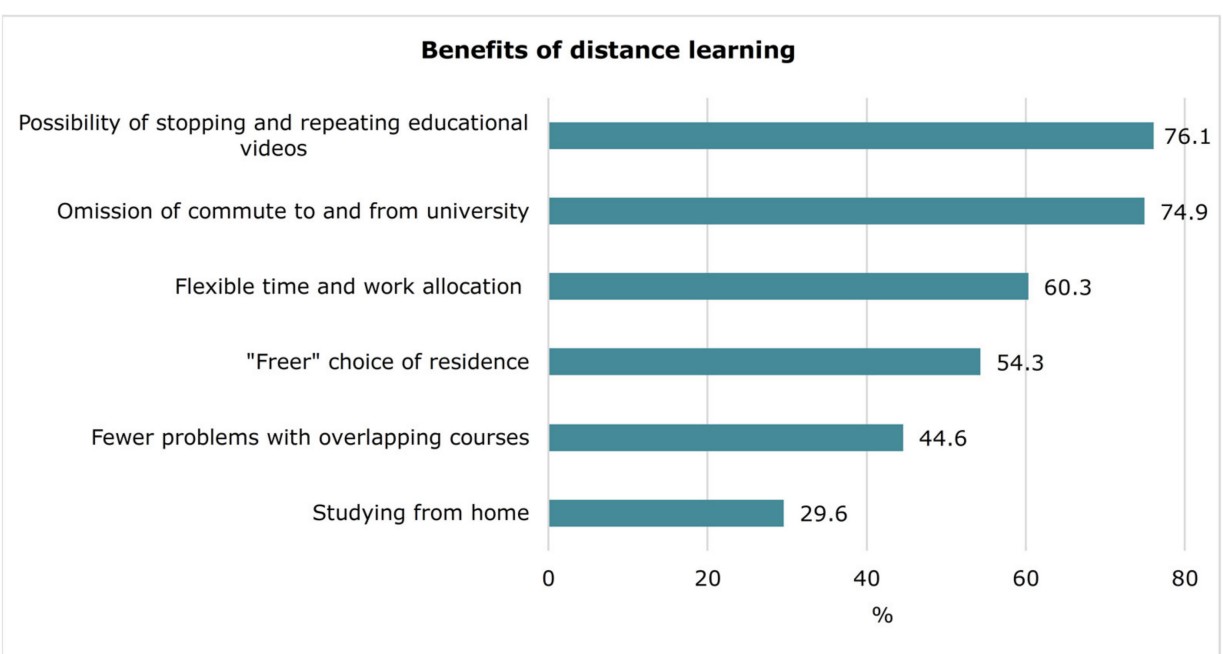

**Figure 5.** Advantages of distance learning that students reported during the winter term 2020/2021 (multiple choice, source: quantitative survey, February 2021).

The students who experienced advantages of distance learning were more satisfied with the situation and reported a better performance during the winter term 2020/21 (see Table 2).

**Table 2.** Group differences between students experiencing advantages of distance learning and students not experiencing them.

| | *n* | M | SD | *n* | M | SD | T | d | *p* |
|---|---|---|---|---|---|---|---|---|---|
| | **Advantage Experienced** | | | **Not Experienced** | | | | | |
| **Satisfaction with Distance Learning** | | | | | | | | | |
| Possibility of studying at one's own pace | 2067 | 3.54 | 0.79 | 645 | 3.18 | 0.86 | 9.58 | 0.45 | <0.001 |
| Omission of commute | 2028 | 3.56 | 0.81 | 684 | 3.14 | 0.77 | 11.80 | 0.53 | <0.001 |
| Flexible time and work allocation | 1637 | 3.67 | 0.76 | 1075 | 3.13 | 0.81 | 17.63 | 0.69 | <0.001 |
| "Freer" choice of residence | 1474 | 3.61 | 0.80 | 1238 | 3.27 | 0.81 | 10.76 | 0.42 | <0.001 |
| Fewer problems with overlapping courses | 1206 | 3.70 | 0.75 | 1506 | 3.26 | 0.82 | 14.73 | 0.56 | <0.001 |
| Studying solely from home | 802 | 3.78 | 0.79 | 1910 | 3.32 | 0.80 | 13.94 | 0.58 | <0.001 |
| **Performance during Distance Learning** | | | | | | | | | |
| Possibility of studying at one's own pace | 1656 | −0.02 | 0.59 | 555 | −0.14 | 0.60 | 4.18 | 0.20 | <0.001 |
| Omission of commute | 1659 | −0.00 | 0.59 | 552 | −0.19 | 0.60 | 6.28 | 0.32 | <0.001 |
| Flexible time and work allocation | 1311 | 0.06 | 0.58 | 900 | −0.21 | 0.58 | 10.85 | 0.47 | <0.001 |
| "Freer" choice of residence | 1185 | 0.01 | 0.58 | 1026 | −0.12 | 0.60 | 5.01 | 0.22 | <0.001 |
| Fewer problems with overlapping courses | 971 | 0.08 | 0.58 | 1240 | −0.15 | 0.59 | 9.26 | 0.39 | <0.001 |
| Studying solely from home | 647 | 0.11 | 0.61 | 1564 | −0.11 | 0.57 | -8.10 | 0.38 | <0.001 |

Note. *n* = sample number for each response option, M = mean, SD = standard deviation, T = *t*-test, d = Cohen's d, *p* = significance level.

The digital accessibility of recordings and videos of lectures was found to be very useful by many students. Many also liked the option of being able to stop and repeat videos for a better understanding of the content. The qualitative data revealed that this accessibility of recordings also enabled the students to organize their learning phases independently of the lecture times, which was appreciated, among others, by working students and students with children:

> *"I see opportunities (...) in the fact that it can be a relief for students who still have to work on the side, because a freer time management is possible. Also, you are very flexible where you are and so you can study from anywhere."* (Elena, 21, Narrative, DC-4)

In the quantitative data there was no significant effect due to children being present in the same household on satisfaction ($t(2678) = 1.18$, $p = 0.237$, $n = 2680$) or performance ($t(229.39) = 0.76$, $p = 0.450$, $n = 2185$). In the interpretation of this result, closures of childcare services should be factored in.

Other students appreciated recorded courses because they facilitated distributing daily screen time throughout the week and helped them avoid packed distance learning days. In addition, others said that distance learning allowed them to take multiple courses with the same or overlapping course times, making it possible for them to study somewhat quicker than they would have in the regular face-to-face teaching system. Especially for students with multiple study subjects this can be beneficial. Students studying more than one subject reported increased performance during the distance learning semester (M = 0.09, SD = 0.60, $n = 495$) compared to students with only one subject, who reported decreased performance (M = −0.09, SD = 0.58, $n = 1602$, $t(2095) = 5.97$, $p < 0.001$, d = 0.31). There was no difference in satisfaction ($t(2567) = 0.33$, $p = 0.742$, $n = 2569$). The degree aspired to (Bachelor, Master, Diploma, teacher, PhD/Doctorate) did not have significant effects on satisfaction with distance learning ($F(4, 2691) = 2.14$, $p = 0.074$, $n = 2695$). There was a significant effect on performance ($F(4, 228.43) = 9.39$, $p < 0.001$, Welch-test, $n = 2201$). Students training to be teachers ($n = 194$) reported a stronger increase in their performance during distance learning than Bachelor, Master, or Diploma students (all $p < 0.001$).

Since physical attendance at the university campus was no longer required or possible, some students chose to leave Innsbruck and return to their home countries/towns and/or parental homes (cf. Section 4.3). Many participants reported that the elimination of necessary travel and commuting resulted in noticeable time savings, a reduction in travel costs, and their attendance at more classes:

> *04/23/20, 11:24—Rebecca: What I like is the online lectures. This allows me to attend significantly more lectures because I have less time issues with driving back and forth. I have lectures on Engineering, Botany, and at [the] Geiwi [campus]—it's really hard to be on time a lot of the time.* (Rebecca, 26, MIMI chat log, DC-2)

While some students, especially at the beginning of the pandemic, struggled with not having work or internships, income from work, or having to make dual conversions from study and internships or work to digital formats, others also appreciated a better work or internship/study balance through distance learning:

> *05.04.20, 11:21—Lucy: Honestly, there are only advantages for me with the current situation in terms of university. Since I work full-time as a teacher and do my studies virtually on the side, I have more leeway to arrange my tasks. The professors are also more relaxed with the submissions and I find that very pleasant. I also like the fact that you don't have to be present at the seminars and that more emphasis is placed on personal responsibility.* (Lucy, 27, MIMI chat log, DC-1)

In this context, some students spoke of increased "convenience" in their studies. Students also rated the experience gained with the home-office work model, and their familiarization with useful learning, working, and communication software, as potentially valuable experiences for the future. Some students found these new opportunities a relief in principle.

Creative and interactive teaching formats and experiences with new digital tools also received particularly positive feedback in the narratives and MIMIs. After all, 49.1% of the students enjoyed using more digital media for distance learning, in comparison to 32.9% who disliked the increased use of digital media. Those who appraised the increased use of digital media as positive reported a higher satisfaction with distance learning (M = 3.70, SD = 0.76, $n$ = 1335) than those who appraised it as negative (M = 3.09, SD = 0.79, $n$ = 891; $t(2224)$ = 18.32, $p < 0.001$, d = 0.79). Additionally, approval of the increased media use was associated with increased performance (M = 0.04, SD = 0.60, $n$ = 1086) but disapproval was associated with decreased performance (M = $-0.17$, SD = 0.57, $n$ = 749, $t(1833)$ = 7.49, $p < 0.001$, d = 0.36).

The participants in the qualitative longitudinal part also noted that they could see a positive development in distance learning over the course of 2020 and that both they and the teachers had slowly adapted or become accustomed to distance learning. Even though many of the challenges and problems in distance education (mentioned above) could not be completely eliminated in the winter term 2020/21, in the quantitative survey, 37.2% found that distance learning had improved slightly and 19.1% found it had improved greatly in comparison to the first semester afflicted by COVID-19, i.e., the summer term 2020.

### 4.4. Places of Stay and Socio-Material Educational Spaces during the Pandemic

At the core of the COV-IDENTITIES project is the study of students' everyday spaces during the pandemic. A variety of factors influenced the home and educational spaces beyond the technical challenges already described in Section 4.3, and thus also students' learning conditions. These include living arrangements, household structures, mobilities and changes of residence during the pandemic, and the related social and material learning environments and the changes thereof.

Looking at the socio-material home spaces, when asked with whom they lived together at the time of the quantitative survey in early 2021, 40.2% reported that they lived with their parents, 23.1% in shared apartments with flatmates, 19.6% with their partners, 8.8% alone, and 4.9% in student accommodation. Combinations of answers were noted in the "other" section (e.g., living together with partners and flatmates in shared apartments). Students living alone in particular reported suffering due to isolation and stay-at-home-orders—especially during the first lockdown (DC-1 and DC-2). At the beginning of 2021, after nearly one year of pandemic and distance learning, the living situation still had significant effects on satisfaction with distance learning ($F(4, 625.16)$ = 10.02, $p < 0.001$,

Welch corrected, $n = 2620$). In particular, students living in shared apartments reported significantly lower satisfaction than all other students ($p < 0.001$ for living with partner or parents, $p = 0.031$ for living alone, Games–Howell corrected). There was no significant effect of the living situation on performance ($F(4, 2133) = 2.00$, $p = 0.093$, $n = 2138$).

At the same time, the qualitative and quantitative data indicate that there has been some flux in places of stay. In this context, the qualitative study points to unusually frequent, mostly temporary, changes of residence as well as, in some cases, to relocations, which are related to the pandemic. Overall, the quantitative survey revealed that 24.4% changed their places of residence once and 13.5% several times during the first year of the pandemic, while 62% did not relocate. There were no effects of relocation on satisfaction with ($t(2334.98) = 1.51$, $p = 0.132$, $n = 2712$) or performance during distance learning ($t(2209) = -0.52$, $p = 0.606$, $n = 2211$). Some of those who relocated decided to leave Innsbruck to be with their families. The qualitative longitudinal study shows that the study participants decided to return to their families especially during the lockdown in spring 2020 and the subsequent summer semester 2020 (in the survey DC-1, DC-2, already significant decrease in DC-3). A smaller proportion of the qualitative study participants also spent the second lockdown in autumn 2020 with families. Almost all students who made this decision describe positive psycho-social effects from returning to their families, especially in DC-1 and DC-2. They also emphasize the availability of recreational and green spaces, especially gardens, but also positive effects of an often-rural location that allows easier access to nature during lockdowns compared to the city:

> "*After 6 weeks of staying in Innsbruck, I decided to start my journey home and go back to the Allgäu [region in Southern Germany] to my family. [...] I really enjoy the exchange of ideas, cooking and eating together with my family and playing with my nephew and niece in the big garden. Even though I have to accept a two-week quarantine again when crossing the border, I find it much easier under these circumstances.*" (Theo, 28, narrative, DC-2)

Rarely, however, negative aspects of moving back in with the family are also described, such as the resurgence of family conflicts, sometimes with accompanying (re)negotiation of parent–child or sibling relationships, and a lack of peace and quiet for learning. We additionally asked if people felt they had a stable circle of friends in Innsbruck, the city of their studies. There were no effects of the stable circle of friends on satisfaction with distance learning ($t(2639) = 0.53$, $p = 0.594$, $n = 2641$) or on performance ($t(2152) = -1.25$, $p = 0.212$, $n = 2154$).

Other study participants reported psychologically stressful socially conflictive phases during the pandemic, which ultimately also had an impact on their learning (see also Section 4.3). These include conflicts with flatmates in shared apartments and with partners—in the case of the latter—both due to spatial separation and to excessive spatial proximity, especially during the lockdowns. These conflicting social situations were for some students also the reasons they moved back to their families, left or changed flat-sharing communities, moved in with their partners, or moved away from them. However, there are also examples in the narratives that emphasize the mutual support between students that some experienced in shared living arrangements:

> "*I get along quite well with distance learning, but also only because my two flatmates study the same as I do and so we can do a lot together and support and motivate each other. I think without the two of them it would be much harder for me to find the motivation to stick with it. However, it's already working out much better this term than last summer term.*" (Alicia, 20, narrative, DC-4)

Spatial confinement, lack of physical space for learning, and lack of sufficiently quiet learning environments are other factors that negatively influenced distance learning (see also Figure 6):

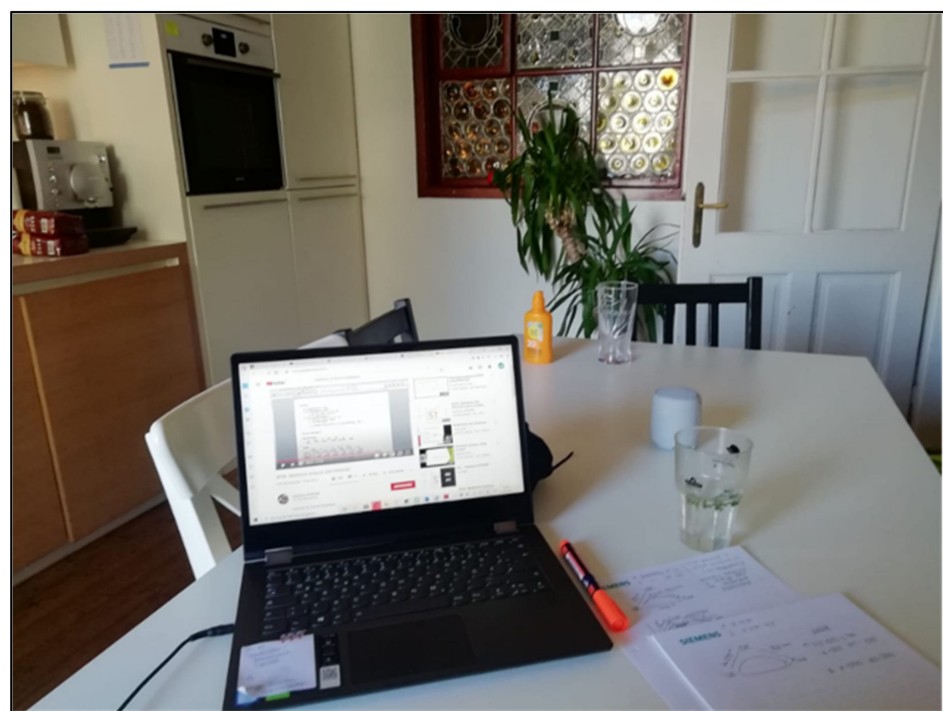

**Figure 6.** Snapshot taken by participant during a MIMI: In the pandemic, the master's thesis is written in the shared kitchen instead of the university library (Stella, 25, MIMI material, DC-2).

> *"However, I do notice how it stresses me out when my flatmates are also here sometimes and both do their distance learning in the kitchen-living room of our shared apartment. Our rooms are very small (about 10 square meters) so there's not really enough room for a desk."* (Aaron, 26, Narrative, DC-4)

The lack of work and study space in the university premises, especially the library, is highlighted frequently in the narratives and MIMIs:

> *23.04.20, 13:00—Interviewer: Does the location (now at home in the home office) change your activity?*

> *23.04.20, 13:01—Alma: Yes, I am unfortunately not as focused at home as I would be otherwise when I am working at the library. And I don't have new books available. Many online libraries are overloaded or not working properly.*

> *23.04.20, 13:03—Interviewer: What distracts you most from your tasks at home?*

> *23.04.20, 13:10—Alma: My consoles (XBOX and Nintendo Switch) and my TV. Short breaks become longer breaks as a result.* (Alma, 25, MIMI chat log, DC-2)

However, the quantitative survey, which asked about a change of the location of study before and after the pandemic, showed that the increase in students who worked at home, at 21.1%, was not as large as might have been expected (see Figure 7). This reflects that many students were also learning at home during normal times. Still, those who had previously used alternative places of study (informal physical spaces of learning), such as a library, friends' places, or when travelling prior to the pandemic, were no longer able to do so during the pandemic and the related restrictions on the use of public learning spaces. Students who reported changes in their location of study ($n$ = 1285, M = 3.30, SD = 0.81) were less satisfied with the distance learning situation than those who did not ($n$ = 1427, M = 3.60, SD = 0.80, $t(2710)$ = 9.64, $p < 0.001$, $d$ = 0.37). Additionally, their performance decreased during the distance learning ($n$ = 1133, M = $-0.12$, SD = 0.60), while the performance of students whose learning location did not change stayed constant ($n$ = 1078, M = 0.03, SD = 0.58, $t(2208.89)$ = 5.80, $p < 0.001$, d = 0.25).

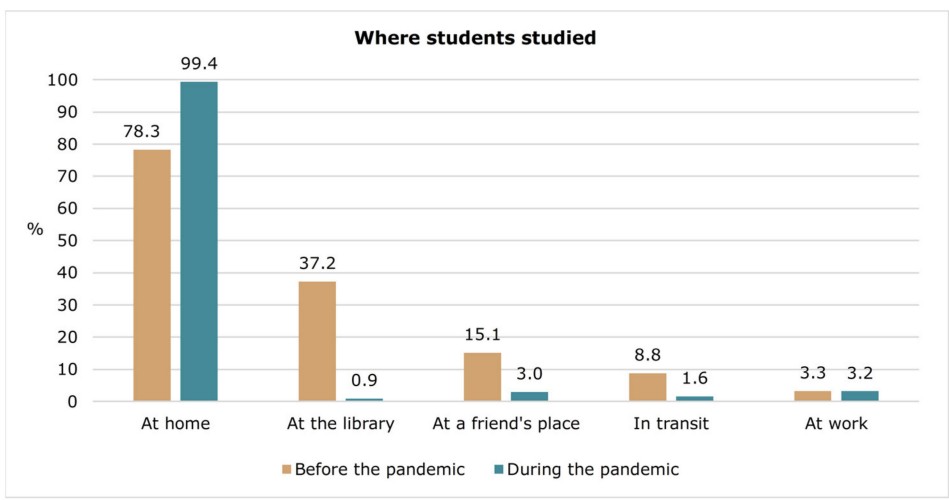

**Figure 7.** Locations of study before and during the pandemic as percentages (multiple choice, source: quantitative survey, February 2021).

### 4.5. Techno-Social Spaces and Polymedia Use during the Pandemic

Everyday spaces of students are characterized by an entanglement of techno-social and socio-material contexts (see Section 2). The qualitative part of the project sheds light not only on the use of digital media for distance learning but also on their further use for private purposes in the broadest sense. This further use of digital media is still highly relevant to distance learning because, on the one hand, it has made a central positive contribution—especially psycho-social, but also in terms of physical health, income, leisure time, civic engagement—as reported by participants in the qualitative study part. On the other hand, the MIMIs, in which students' everyday practices were surveyed for a full day at each survey phase, illustrate that distance learning was very often accompanied by private, parallel media use. Some students used several programs simultaneously, often on multiple screens and devices. With their polymedia concept, Madianou and Miller [52] point out how such converged media use has various social and emotional consequences for the users.

The following WordCloud illustrates the diversified use of digital media (see Figure 8). The WordCloud immediately highlights the central role that social media played for the participants—social media such as WhatsApp, Instagram, Skype, and YouTube were *also* used in distance learning, but primarily for private purposes.

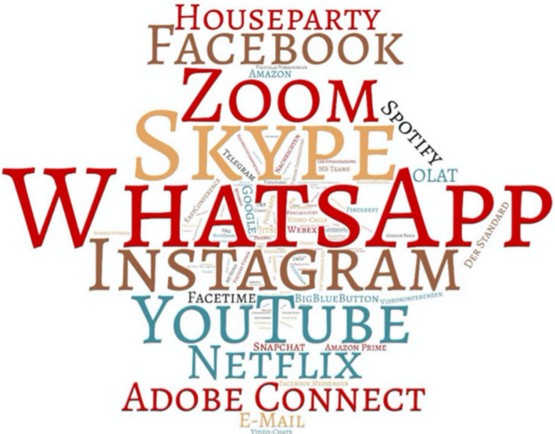

**Figure 8.** WordCloud of software used by students for private purposes and distance learning during the pandemic (visualized are all instances of software named in the narratives; COV-IDENTITIES project).

The resulting blurring of private and learning contexts, of university and home, caused more trouble for some students in the context of distance learning, promoting stress and the feeling of being overwhelmed, see [38]:

> "*I do everything university-related in the living room, so now I will always associate the living room with university.*" (Lukas, 20, Narrative, DC-3)

### 4.6. Students' Aspirations and Recommendations for Distance Learning

In the qualitative narratives and interviews, students were asked about their aspirations and recommendations for making distance learning more enjoyable and productive. These were then quantified by letting the participants of the quantitative survey choose of which recommendations and aspirations they approved. Many of the answers pointed to elements of teaching, which were not only relevant during a pandemic but also beyond (see also Figure 9).

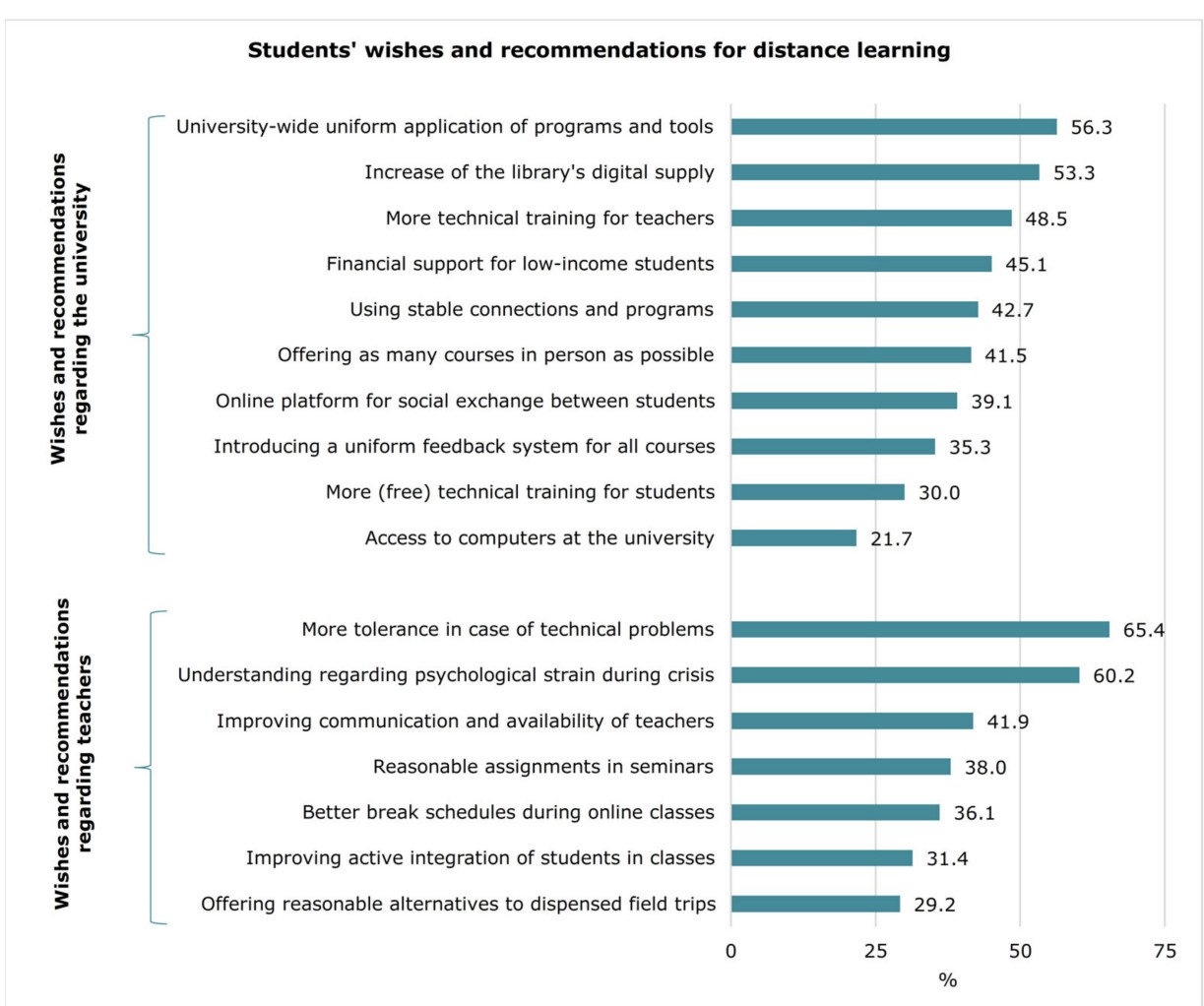

**Figure 9.** Students' wishes and recommendations regarding digital media for distance learning (multiple choice, source: quantitative survey, February 2021).

Despite the advantages that distance learning could have for them, students wanted to experience as much face-to-face teaching as possible during the pandemic situation. In addition, many students expressed the desire for adequate alternatives for field and laboratory experiences, such as field trips and practical training sessions, which would be cancelled due to the pandemic. Pure assignment processing instead of practical exercises and interactive teaching formats was not seen as a suitable and acceptable substitute. Fur-

thermore, a majority of students wanted to see a continuation and expansion of the practice of recording courses and making instructional videos available to students. Also relevant for regular teaching were the expressed wishes for more online courses, more blended learning formats, and the introduction of an evaluation or feedback system that accompanies the teaching period in order to allow teachers to implement possible improvements in course quality while courses are still running. With regard to the design of (distance) learning courses, students also wished for more interactivity, more room for discussion, a reduction of computer-based assignment processing, and the aforementioned avoidance of assignment processing as an alternative to face-to-face courses. The desire for more breaks was also frequently expressed. In the quantitative survey, 56.2% reported that not taking (enough) breaks was an issue for them. Ronja emphasized their importance in a qualitative interview:

*"Breaks during seminars are very important. [laughs] Oh my God, that's a huge thing that a lot of people forget when they're sitting in front of the computer, that you need breaks because you just don't have anyone to give you direct feedback. Breaks! Incredibly important! I think it's even worse to listen in front of the computer than anywhere else. That means taking a short break every half hour, preferably every 45 min, so that you can just take a breath, get fresh air, etc."* (Ronja, 27, interview, November 2020)

In the qualitative data collection, lecturers were asked for more reachability, more commitment and creativity, consideration, and tolerance for students with technical problems, as well as clearly formulated and comprehensible assignments. In addition, the desire for obligatory (technical) training for teachers was mentioned in order to compensate for what students perceived as a lack of technical competence on the part of some teachers. All these wishes are also directed at regular teaching and are also formulated in the following narrative excerpt by Paul (24, DC-4):

*"Lecturers [should] be encouraged to be more creative in online teaching. The frontal teaching without interactive elements by technically limited lecturers, which is often propagated as being without alternative, could be loosened up by training, technical support (possibly students with an affinity for technology) and best-practice examples. Intermedia formats are already being used successfully by a few lecturers."*

The universities and institutes were asked to continue with a continuous flow of information about important innovations, changes, and developments in connection with teaching. In addition, students would like to see uniform pandemic-related regulations across institutes with regard to classroom teaching, distance learning, alternatives for internships and excursions, and the use of software.

For example, some students wanted more access to (technical) training for themselves and their instructors in order to better master the challenges that come with (distance) learning. In addition, there was a desire for further development of software so that it both offers more features and is less error-prone due to so-called bugs (errors in the software code/operation). Additionally, free access to course-relevant software would address some of the difficulties students face. A continuation of digital formats, as well as the use of uniform software across institute borders, is just as urgently recommended as is the rental of hardware or financial subsidies from the university to students for necessary hardware purchases. Especially during a pandemic, students who do not have the necessary hardware would appreciate having access to powerful computers either physically in the university building or digitally via remote access from home.

In the quantitative survey, students were asked to choose elements of distance learning which they would like to keep even after the pandemic (see Figure 10), with a large majority stating their preference for maintaining online access to all materials—since this question did not inquire as to how many perceived they already had good access prior to the pandemic, it is not certain how much change students would like here. Significantly, a high percentage of students wanted to keep the recording of classes, and slightly more than half were interested in maintaining opportunities for distance learning overall.

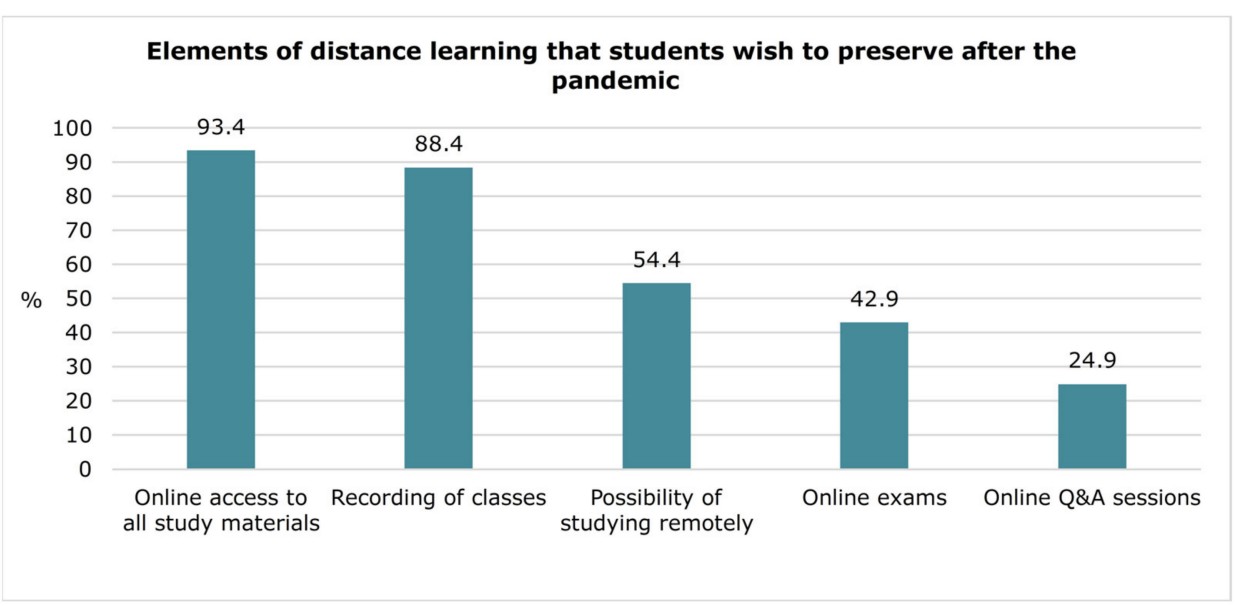

**Figure 10.** Elements of distance learning that students wish to maintain even after a return to presence teaching (multiple choice, source: quantitative survey, February 2021).

## 5. Discussion

Despite the challenges perceived by students during the past year of pandemic education, the current generation of students is able to continue with many elements of their studies through the opportunities that distance learning offers. A large proportion of the students in our study have managed to continue and even successfully complete their university education in this past year characterized by the COVID-19 pandemic. Nevertheless, it is probably more appropriate to speak of them as persevering during this time. For them, education during the crisis period meant being much more self-reliant and having to deal very flexibly with a potpourri of different virtual-teaching formats.

In terms of pandemic education, our research participants reported a significant improvement since the beginning of the pandemic in spring 2020. Especially at the beginning of the pandemic, the change often meant self-study—lists of teaching materials and literature were sent out, teaching staff first had to get used to virtual-teaching formats, and the software used caused difficulties. The satisfaction with distance learning improved with both teachers and learners adjusting to the distance learning formats.

Through the evaluation of the qualitative and quantitative data, moreover, it became clearly visible that the students of the Innsbruck universities were satisfied with the information provided by the universities via websites and social media, and that the information and communication skills of the universities were rated positively. Aristovnik et al. [13] state in their study that students worldwide rate their respective universities with similar degrees of positivity in this regard. This points to the importance of efficient communication on the part of university management and argues for maintaining a communication strategy that keeps students informed about developments at universities.

Nonetheless, as in other countries e.g., [53–57], the students in our Austrian study perceived a variety of challenges. Many participants urgently desired a return to face-to-face teaching and university life as well as direct face-to-face contact with teachers, their classmates, and other fellow students. The limitations of adequately replacing practical, field and laboratory courses during conditions of distance learning were strongly criticized. For some, technical difficulties posed a barrier, especially poor or overloaded internet connections and insufficiently powerful computers. Restless learning environments and the feeling of not being able to escape the constant virtual presence of university in the private room were also among the everyday realities of the pandemic. Young people were confronted with many uncertainties during this time, while contact persons were

often less tangible than in normal times. This widening of inequalities because of the pandemic promises to be particularly problematic and poses an impediment to reaching the Sustainable Development Goal of quality education for all [29,58]; those students who do not have sufficient psycho-social support from families, who have no (adequate) learning space at home, who lack the financial means (e.g., for access to hardware and software), and those who have already suffered from psychological stress and pre-existing conditions are left further behind. The psychological effects of the pandemic and distance learning on students described in this study, especially the increased risk of depression, are in line with the study results of Huckins et al. [10] (see also [59]). Accordingly, students wish for more understanding of mental health problems during this crisis.

Another problematic aspect of the current distance learning relates to difficulties in time management as the shift to distance learning not only leads to spatial changes, but also to a lack of institutionally framed time management [34]. Students reported problems in taking adequate breaks and in achieving a time structure while learning from their homes. This suggests they need more support through both having temporal frameworks set by their teachers or universities, and education on how to improve at self-management and setting up their own time structures.

However, most students clearly would like to see the continuation and, in most cases, significant further development of those gains from digitization from the first pandemic year: blended learning formats, the widespread use of online learning-management systems, and the use of creative digital tools [29]. Strikingly, in the quantitative survey, even more than half of the participants expressed their wish to maintain the option of overall distance education even after the pandemic. The primary reasons are, especially, greater flexibility, time savings, and more options to adjust learning to one's own speed and abilities; new forms of educational and learning spaces seem to empower students when organizing their own learning. Especially students who work during their studies or who have children to care for seem to profit from the increased flexibility, which thus can help lessen social inequalities and make university education more accessible. Based on the perspectives of students, it is appropriate to demand a significant change in post-pandemic education. After all, since the switch to online formats affected not only the education but also the work sectors, it seems that enhanced digitization of university education is urgently needed to prepare graduates for the realities of a working life in the 21st century.

## 6. Conclusions and Recommendations

Although the advent of the "post-digital" [2]—an era in which the digital has become a natural, banal, and ubiquitous part of everyday life—has long since been declared, many universities in Austria and beyond have (knowingly) been lagging behind in adapting education to these times. This condition has played out painfully for students, teachers, and eventually society during the COVID-19 pandemic. Based on the results of our mixed-methods study we conclude by pointing out recommendations for action in dealing with this situation during continued higher education during the pandemic. We also sketch pathways for post-pandemic educational futures that are socially more sustainable in preparing young people for their lives in the post-digital.

In the face of an ongoing and uncertain pandemic situation, the following short-term recommendations for lecturers are aimed at helping to improve remote-teaching processes in order to support students in dealing with managing their university education during these challenging times [29,60]. Lecturers are recommended, where possible, to reduce computer-based assignment processing, insert more frequent breaks in classes (at least every 30–45 min), and to make teaching more interactive as well as involve students more closely. Particularly, lectures can be recorded and provided online to provide more time and spatial flexibility to students. Students' parallel media use for private purposes poses an increasing challenge during online classes—just as during presence teaching, when students bring their mobile devices to class. Our results indicate, however, that this challenge has increased all the more during distance learning. As

measures aimed at reducing private media usage during distance learning, we recommend using interactive and changing teaching formats, establishing and reminding students of netiquette, raising students' awareness of negative effects on mental health and learning outcomes, and encouraging the use of the camera function (if necessary, temporarily). The loss of opportunities for in-class contact could be partially mitigated through offering students to meet teachers in the virtual classroom before or after each class and the extension of online consultation hours. To support contact between students, teachers are further encouraged to implement teaching methods that promote interaction. Even though group work is perceived as being more difficult to conduct digitally, group assignments can still provide a first step toward creating closer contacts between students.

While lecturers are the first point of contact for most students during distance learning, universities as institutions can and should help to support all parties involved based on the following short-term recommendations for action for general teaching. University management and administration are recommended to educate students and lecturers on health-promoting measures in the home office to reduce physical discomfort and prevent fatigue and stress, which is detrimental to quality learning [28,61]. Both students and lecturers need to be made more aware of the physical and mental health strains of distance teaching and learning, so as to increase their understanding and tolerance of the shortcomings of others. The use of standardized software in teaching across administrative-unit boundaries needs to be fostered. Universities need to find solutions to provide adequate alternatives to field, laboratory, and practical exercises. If this is not possible, they are recommended to provide intensive catch-up formats for such classes once attendance teaching resumes. To support social exchange between (particularly first-year) students, (a) student online platform(s) for exchange and/or joint learning should be established. Admission to these platforms should be made subject to prior agreement to the adherence to netiquette in these platforms. Universities should make teachers aware of restrictions that individual students potentially face during the pandemic resulting from socio-material learning environments, mental stress, and technical challenges. Exceptions for individual students should be permitted and developed to ensure accessibility to education for disadvantaged students. Resumption of face-to-face teaching as soon as legally—and from a public-health perspective—possible again, should be announced with sufficient advance notice to allow students who have returned to live with their families to come back to the universities' locations and rent accommodation in time.

With advancing vaccination rates, post-pandemic educational futures become more tangible, and the need for empirically well-grounded discussions on such futures increases. Based on our extensive study, we see the implementation of the following medium-term measures as essential to make teaching future-proof and distance learning more learner-centered and sustainable beyond the phase of the pandemic [62]: the continuation of the digitization of university processes, literature, and library services must be strongly fostered. Based on this study, the retention of online course offerings, the establishment of hybrid courses, and the strong encouragement of creative and innovative online and blended teaching methods are recommended [27]. Current distance learning formats should not be adopted unchanged, but prevailing problems should be removed, such as the lack of social contacts and time structure. The technical possibilities of (new) learning software are continuously developing and expanding. They offer new opportunities but also present challenges for lecturers. Regular and compulsory training for teachers in these new opportunities is, in our opinion, the only option to keep university teaching up-to-date with the young generations' expectations of education in the post-pandemic, post-digital age. For teachers to be able to reserve sufficient time to acquire and update their digital skills, it will be necessary for universities to offer corresponding teaching reductions. After all, the acquisition of digital skills is an important pillar of tertiary education, preparing students for an increasingly digitized labor market and society. As has been long noted by (critical) education scholars, pupils and students need skills that prepare them for a responsible and active use of digital media, allowing them to become mature digital

participants in society (e.g., [63,64]). This is an essential component of socially sustainable education and society in the post-digital.

**Author Contributions:** Conceptualization: T.B.-H., B.J., M.H.W., F.B. and K.K.; methodology: T.B.-H., K.K., F.B., D.M.I., B.J. and M.H.W.; validation: T.B.-H. and V.K.; formal analysis: F.B., T.B.-H., V.K. and D.M.I.; investigation: T.B.-H., F.B., V.K., D.M.I. and A.M.; resources: T.B.-H., B.J. and M.H.W.; data curation: T.B.-H. and V.K.; writing—original draft preparation: T.B.-H., F.B., V.K., K.K. and A.M.; writing—review and editing: K.K., T.B.-H., V.K., A.M. and F.B.; visualization: D.M.I., V.K., F.B. and T.B.-H.; Supervision: T.B.-H., B.J. and M.H.W.; project administration: T.B.-H., B.J. and M.H.W.; funding acquisition: T.B.-H., B.J. and M.H.W. All authors have read and agreed to the published version of the manuscript.

**Funding:** This research was funded by the University of Innsbruck, project number 338221, and by Förderkreis 1669—WissenschafftGesellschaft, grant number 329327.

**Institutional Review Board Statement:** Ethical clearance was provided for the COV-IDENTITIES project by the Board for Ethical Questions in Science of the University of Innsbruck, Innsbruck, Austria, on 1 April 2020.

**Informed Consent Statement:** Informed consent was obtained from all subjects involved in the COV-IDENTITIES study.

**Data Availability Statement:** The quantitative data are not yet publicly available, as the data analysis and processing are still ongoing. The qualitative data are not publicly available due to privacy and ethical reasons.

**Acknowledgments:** The authors thank the student participants and the student researchers involved in this project. Further, the authors thank the anonymous reviewers as well as the Open Access Fonds of the University of Innsbruck for their generous support of the publication of this article.

**Conflicts of Interest:** The authors declare no conflict of interest.

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
