# Peer review of "University Students’ Perception, Evaluation, and Spaces of Distance Learning during the COVID-19 Pandemic in Austria: What Can We Learn for Post-Pandemic Educational Futures?"

_sustainability, doi:10.3390/su13147595_

Round 1

Reviewer 1 Report

This is a well done study, timely, needed, and stands to make a great contribution to COVID-19 higher education research/literature. A few things to improve:

  • the qualitative methods section could use more depth- explain how the data were coded (by hand? through software? what were the researchers' roles?)
  • some of the qualitative data procedures are in the results -such as how the word clouds were developed - move any methods descriptions from the results to the methods section
  • the results section is very dense and somewhat tricky to follow with the weaving together of the quantitative and qualitative data - the authors should be commended for weaving together the data sources to narratively explain the students' experiences - however, spend some time in the methods section to explain how the data sources were woven together and approach to the results
  • what steps were taken for trustworthiness and credibility of the study? did the participants member-check the results?
  • watch out for biased and loaded language in the results such as "complain" - that's a researcher interpretation of the participants' tone - consider instead using words such as described, said, explained, etc. 

Overall, well done study and much needed in education research.

Reviewer 2 Report

It is an up-to date topic, interesting, but very many tables. Mabye they could be reduced, and with that also cut out some of the results, since the article is very long. My main question is how is this topic and the results are related to sustabaility questions. That is something I would like to see a discussion about, now teh authors in a way take it for granted that is is, but it has to be argued for and explained in what we this is sustainability from your perpective.

Reviewer 3 Report

The article is about  students’ sprception, evaluation and spaces of distance learning during the COVID-19 pandemic. However, some aspects must to be revised:

  1. The title of section 2 must to be revised or deleted, due to the fact that the contents of this section can be introduced in section 1 Introduction, but briefly described.
  2. For highlighting the importance of the research, fig 1 and fig 4 can be converted in only one!
  3. Particular comments of the participants and examples from sections 4.3.1. and 4.3.2. or 4.4. (Rebecca, Anika, Elena, Alma, etc.) can be erased: has no scientific soundness.
  4. Picture 1 – not relevant.
  5. Figure 7 show something very strange: students don’t study in library because is not allowed during pandemic, not because they don’t want anymore: this is a false conclusion/result. If the authors will compare “before pandemic” with “after pandemic” then the result can be significant. Other results in figure 7 is not relevant.
  6. Please erase figure 8. No scientific meaning!
  7. Please refine conclusions: too expansive and hard to understand!
  8. Please use numbers for easy identifying the references in the text af the article.

Round 2

Reviewer 3 Report

The comments were completed by the authors.